# Correlates of variability in endurance shuttle walk test time in patients with chronic obstructive pulmonary disease

Anouk A. F. Stoffels[1,2,3]*, Bram van den Borst[1], Jeannette B. Peters[4], Mariska P. M. Klaassen[1], Hanneke A. C. van Helvoort[1], Roy Meys[2,3], Peter Klijn[5,6], Chris Burtin[7], Frits M. E. Franssen[2,3], Alex J. van 't Hul[1], Martijn A. Spruit[2,3], Hieronymus W. H. van Hees[1], on behalf of the BASES consortium[¶]

1 Department of Pulmonary Diseases, Dekkerswald Radboud University Medical Center, Radboud Institute for Health Sciences, Nijmegen, The Netherlands, 2 Department of Research and Development, CIRO, Horn, The Netherlands, 3 Department of Respiratory Medicine, Maastricht University Medical Center (MUMC+), NUTRIM School of Nutrition and Translational Research in Metabolism, Maastricht, The Netherlands, 4 Department of Medical Psychology, Radboud University Medical Center, Radboud Institute for Health Sciences, Nijmegen, The Netherlands, 5 Department of Pulmonary Rehabilitation, Merem Medical Rehabilitation, Hilversum, The Netherlands, 6 Department of Pulmonary Medicine, Amsterdam University Medical Center, Amsterdam, The Netherlands, 7 Reval Rehabilitation Research–Biomedical Research Institute, Faculty of Rehabilitation Sciences, Hasselt University, Diepenbeek, Belgium

¶ Membership of the BASES consortium is listed in the Acknowledgments.
* anouk.stoffels@radboudumc.nl

**Data Availability Statement:** The data underlying this study has been uploaded to the EASY DANS

## Abstract

### Background

The endurance shuttle walk test (ESWT) is used to evaluate exercise tolerance in patients with chronic obstructive pulmonary disease (COPD). The recommended pre-intervention tolerated duration (Tlim) is between 3–8 minutes for optimal interpretation of treatment effects. However, this window may be exceeded and factors determining ESWT Tlim are not completely understood. Therefore, we aimed to determine whether pulmonary function, physical and incremental shuttle walk test (ISWT) performance measures are associated with ESWT Tlim in COPD patients.

### Methods

Assessment data from patients eligible for pulmonary rehabilitation was retrospectively analyzed. Inclusion criteria were: diagnosis of COPD and complete data availability regarding ESWT and ISWT. Patients performed an ESWT at 85% of ISWT speed and were divided into three groups (ESWT Tlim: <3 minutes, 3–8 minutes, >8 minutes). Subject characteristics, severity of complaints, pulmonary function, physical capacity and activity, exercise tolerance and quadriceps muscle strength were evaluated.

### Results

245 COPD patients ($FEV_1$ 38 (29–52)% predicted) were included. Median ESWT Tlim was 6.0 (3.7–10.3) minutes, 41 (17%) patients walked <3 minutes and 80 (33%) patients walked

database and can be accessed at the following DOI: https://doi.org/10.17026/dans-xd6-2fw9.

**Funding:** The BASES consortium is financially supported by Lung Foundation, the Netherlands (#5.1.18.232). Dr. F.M.E. Franssen received support in the form of grants and personal fees from AstraZeneca, personal fees from Boehringer Ingelheim, personal fees from Chiesi, personal fees from GlaxoSmithKline, grants and personal fees from Novartis, personal fees from TEVA, outside the submitted work. Dr. B. van den Borst received support in the form of personal lecture fees from AstraZeneca and Boehringer Ingelheim bv. The funders had no role in study design, data collection and analysis, decision to publish, or preparation of the manuscript. The specific roles of these authors are articulated in the 'author contributions' section.

**Competing interests:** The authors have read the journal's policy and have the following competing interests: F.M.E. Franssen is supported by grants and personal fees from AstraZeneca, personal fees from Boehringer Ingelheim, personal fees from Chiesi, personal fees from GlaxoSmithKline, grants and personal fees from Novartis, personal fees from TEVA, outside the submitted work. B. van den Borst is supported by personal lecture fees from AstraZeneca and Boehringer Ingelheim bv. A.A.F. Stoffels, R. Meys, H.W.H. van Hees, P. Klijn, C. Burtin, M.A. Spruit, H.A.C. van Helvoort, J.B. Peters, M.P.M. Klaassen and A.J. van 't Hul declare that they do not have a conflict of interest. This does not alter our adherence to PLOS ONE policies on sharing data and materials. There are no patents, products in development or marketed products associated with this research to declare.

>8 minutes. Body mass index, maximal oxygen consumption, Tlim on constant work rate cycle test, physical activity level, maximal ISWT speed, dyspnoea Borg score at rest and increase of leg fatigue Borg score during ISWT independently predicted Tlim in multivariate regression analysis ($R^2 = 0.297$, $p < 0.001$).

## Conclusion

This study reported a large variability in ESWT Tlim in COPD patients. Secondly, these results demonstrated that next to maximal ISWT speed, other ISWT performance measures as well as clinical measures of pulmonary function, physical capacity and physical activity were independent determinants of ESWT Tlim. Nevertheless, as these determinants only explained ~30% of the variability, future studies are needed to establish whether additional factors can be used to better adjust individual ESWT pace in order to reduce ESWT Tlim variability.

## Introduction

The endurance shuttle walk test (ESWT) is commonly used to evaluate effects of interventions on exercise tolerance in patients with chronic obstructive pulmonary disease (COPD) [1, 2] in both research and clinical settings [3]. This accessible and low-cost field walking test is performed at an imposed constant pace and is therefore better controlled than other field walking tests, like the 6- and 12-minute walking tests [4–6]. Furthermore, the tolerated duration (Tlim) of the ESWT is considered to be highly responsive to interventions, especially in comparison to maximal walking tests [1, 7] and the change in ESWT Tlim has been associated with change in exercise capacity and quality of life [8, 9].

However, the potential effect size of interventions on Tlim of constant load tests is strongly determined by the load on which the test is performed [10]. Since Tlim has a negative hyperbolical relation with the relative load of the test, testing at higher relative loads will yield less potential improvement on Tlim [10–12]. Accordingly, ESWT load, i.e. pace, is set at a fixed percentage (usually 85%) of the maximum walking pace, pre-determined by an incremental shuttle walk test (ISWT) [6]. Despite this fixed pace, considerable variability in ESWT Tlim was recently observed in patients with COPD (ESWT Tlim = 353 seconds, 95% CI [299–407] [13]). Because effects sizes of interventions on ESWT Tlim depend on pre-intervention ESWT Tlim, a large variability in pre-intervention ESWT Tlim complicates statistical analysis of intervention efficacy and increases the number of participants required in clinical studies [12, 14]. Accordingly, a pre-intervention Tlim between 3 and 8 minutes has been recommended for constant load exercise tests, like the ESWT [15].

It is currently not completely understood why Tlim of some patients falls outside the recommended timeframe of 3–8 minutes. However, we do know that causes of exercise intolerance are multifactorial and heterogenous in patients with COPD [16]. Next to the severity of pulmonary dysfunction, extrapulmonary features, like muscle weakness and psychological status are known to determine tolerance to exercise [10, 16, 17]. Furthermore, the variability of endurance time on a constant work rate cycle exercise test (CWRT) with equal relative loads for all COPD patients was only partly explained by peak exercise capacity and maximal quadriceps strength [18], suggesting that variability of endurance time is determined by additional clinical variables. Whether these factors influence ESWT Tlim as well has not yet been

determined. Lastly, procedural factors can also play a role in ESWT Tlim variability. For example, over- and underestimation of the maximal speed obtained from ISWT can lead to ESWT performance at an intensity not truly representing 85% of the peak capacity. Hence, a performance of the ISWT in accordance with the European Respiratory Society/American Thoracic Society Technical Standards is important [19], as well as maximal effort of the patient.

Collectively, considering that despite protocolized execution of ISWT and ESWT substantial heterogeneity is observed in ESWT Tlim in patients with COPD, hampering clinical evaluation of interventions, there is a need to better understand determinants of ESWT Tlim variation. Therefore, our primary aim was to determine whether pulmonary function, physical and ISWT performance variables are associated with ESWT Tlim in patients with COPD. A priori, we hypothesized that parameters of pulmonary function and physical performance are independent determinants of ESWT Tlim and can partly explain the high variability of ESWT Tlim in patients with COPD.

## Materials and methods

Retrospective analyses were performed on an anonymized dataset from 306 patients that attended a comprehensive pulmonary rehabilitation (PR) program in Dekkerswald–Radboudumc (Nijmegen, The Netherlands) between September 2016 and December 2019. The data was collected during baseline assessment as part of standard care of the PR program. Inclusion criteria for the analyses were a primary diagnosis of COPD according to the Global Initiative for Chronic Obstructive Lung Disease criteria [20] and complete data availability regarding ISWT speed and ESWT speed and time. These criteria were met by 245 patient. A flowchart of in- and exclusion of patients is depicted in Fig 1.

This study was in accordance to the principles of the Declaration of Helsinki. The local ethical board Arnhem/Nijmegen, The Netherlands, informed the authors that the Medical Research Involving Human Subject Act (WMO) did not apply to this retrospective study (2020–6621).

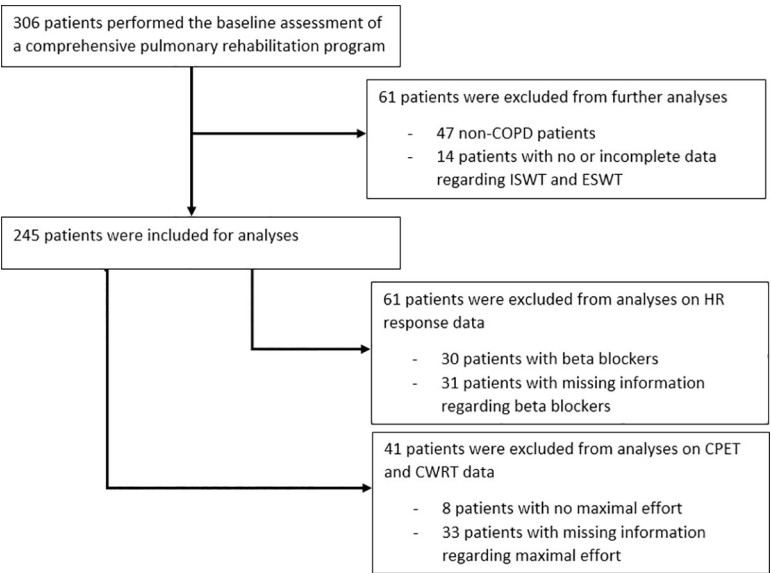

**Fig 1. Flowchart of in- and exclusion of patients for analysis.** Definitions of abbreviations: COPD = chronic obstructive pulmonary disease, CPET = cardiopulmonary exercise test, CWRT = constant work rate test, HR = heartrate, ISWT = incremental shuttle walk test, ESWT = endurance shuttle walk test.

## Measurements

Subject characteristics and severity of complaints as age, gender, weight, body mass index (BMI), Charlson Comorbidity Index [21], fat-free mass index, modified Medical Research Council (mMRC) [22], COPD Assessment Test [23], Hospital Anxiety and Depression Scale [24] and Checklist Individual Strength Fatigue [25] were systemically assessed.

**Pulmonary function tests.** Post-bronchodilator pulmonary function tests including spirometry (forced expiratory volume in one second, $FEV_1$; Tiffeneau index, $FEV_1$/vital capacity), static lung volumes (residual volume, RV; functional residual capacity, FRC; total lung capacity, TLC) and diffusion capacity for carbon monoxide ($DL_{CO}$) by single-breath method (MasterScreen PFT/Body; Jaeger, Würzburg, Germany) were executed according to the European Respiratory Society Recommendations [26] and related to predicted normal values [27, 28].

**Physical performance tests.** The ISWT required the patients to walk around two markers set nine meters apart (10 meters course) at a speed which increases every minute indicated by a pre-recorded audio signal. The patients were instructed to walk for as long as possible [29]. The ESWT was performed at 85% of the maximal ISWT speed and used the same course and auditory signal method. In contrast to the ISWT, the patients were required to walk at a constant speed throughout the test for as long as possible. The ESWT had a maximum test duration of 20 minutes for practical reasons [6]. Both tests were performed according to standardized protocols [6, 29] with on average one week in-between. The following ISWT and ESWT parameters were recorded: Tlim, walking distance, speed, resting and maximal values of transcutaneous peripheral oxygen saturation ($SpO_2$), heartrate (HR) and Borg scores (dyspnoea and leg fatigue). Furthermore, the ISWT distance in meters was calculated as percentage of predicted [30]. Patients with beta blockers (n = 30) or missing information regarding beta blockers (n = 31) were excluded from analyses on HR responses to exercise during both shuttle walk tests (Fig 1).

A symptom-limited ramp maximal cardiopulmonary exercise test (CPET) was performed on an electromagnetically braked cycle ergometer (Ergoselect, Ergoline, Bitz, Germany) according to the recommended guidelines [31] to determine the maximal workload (Wmax) and oxygen uptake ($VO_2$max). Furthermore, the maximal HR was recorded in order to determine the maximal HR during the ISWT relative to the maximal HR during the CPET ($HRmax_{ISWT}$/$HRmax_{CPET}$). The CWRT was performed at 65% of Wmax on the same cycle ergometer as the CPET. Patients cycled until symptom limitation or until pedalling rate decreased under 60 rotations per minute, with a maximum of 20 minutes. Only data of patients that performed CPET with maximal effort, as based on the European Respiratory Society and American Thoracic Society/ American College of Chest Physicians statements on CPET, were included in analysis of CPET and CWRT variables (n = 204) [31, 32] (Fig 1).

Isometric quadriceps strength (maximal voluntary contraction) was assessed with a computerized dynamometer (Biodex System 4 Pro, Biodex Medical Systems, Inc., New York, USA). Participants performed three maximal unilateral isometric knee extensions for five seconds at a knee angle of 60˚, interspersed with 15 seconds of rest. The maximal voluntary contraction was defined as the highest peak torque (Nm) [33] and was both expressed as absolute value as well as related to predicted normal values [34].

Physical activity was measured using the Dynaport MoveMonitor (McRoberts BV, The Hague, The Netherlands) for a duration of seven (with a minimum of at least five) consecutive days and defined as steps per day and average physical activity level (PAL) [35, 36].

## Statistical analysis

Statistical analyses were performed using SPSS statistical software program (IBM, New York, USA), version 25.0. Descriptive data were presented as mean ± SD, median (interquartile

range 25–75%) or number of patients (percentage), as appropriate. Based on the ESWT Tlim, the subjects were divided into three groups (group 1: <3 minutes, group 2: 3–8 minutes, group 3: >8 minutes). These cut-off points were chosen to reflect the desirable ESWT duration of 3 to 8 minutes [15]. Accordingly, in the results section we focussed on differences in groups 1 and 3 compared to group 2.

Between-groups comparisons for continuous variables were tested by one-way analysis of variance (ANOVA) or Kruskal-Wallis test, as appropriate. Categorical variables were tested with a Chi-square test. When a statistically significant difference was obtained, a pairwise post-hoc test was performed and Bonferroni post-hoc testing was applied to correct for multiple comparisons. A p-value of <0.05 was considered significant.

Univariate and multivariate linear regression models were used to evaluate the association of pulmonary function, physical and ISWT performance variables with the ESWT Tlim. Univariate linear regression models were built using the ENTER method. Explanatory variables with a p-value <0.20 and not strongly correlated (r <0.8) with another variable of interest were used to build a multivariate linear regression model, using the backward method. Variables with a p-value <0.05 in the multivariate linear regression model were considered as independent predictors of ESWT Tlim.

## Results

The included patients had a mean age of 61.4±7.8 years, a mean BMI of 25.8±5.7 kg/m$^2$, a median $FEV_1$ of 38 (29–52)% predicted and 47% were male (Table 1). The median ESWT Tlim was 6.0 (3.7–10.3) minutes. A total of 41 (17%) patients walked <3 minutes (group 1), 124 (50%) patients walked between 3–8 minutes (group 2) and 80 (33%) patients walked >8 minutes (group 3). Furthermore, 42 (17%) patients reached the maximum test duration of 20 minutes. The distribution of patients according to the ESWT Tlim is depicted per minute in Fig 2.

### Subgroup characteristics

Gender, BMI, fat free mass index and Charlson Comorbidity Index were similar between groups. Patients in group 3 were younger (59.4±8.6) than patients in group 2 (62.4±7.2, p = 0.021). The severity of dyspnea sensation, as reflected by the mMRC score, was lower in patients from group 3 (median 2(1–3) and mean 1.8±1.2) than group 2 (median 2(1–3) and mean 2.2±1.2, p = 0.006). There were no differences in severity of complaints, COPD Assessment Test, Hospital Anxiety and Depression Scale and Checklist Individual Strength Fatigue scores, between the groups (Table 1).

**Pulmonary function and physical performance parameters.** Pulmonary function of patients in group 2 were similar to patients in group 1 and 3, except for a lower $FEV_1$ (L) in group 1 than group 2 (p<0.001). Measures of physical performance, like maximal exercise capacity, muscle strength and physical activity were comparable between patient in group 1 and 2. Patients in group 3 had a better physical capacity (Wmax and $VO_2$max) and were more physically active (steps/day and average PAL) in comparison to group 2 (all p-values <0.001) (Table 1).

**ISWT performance parameters.** The median ISWT distance of all patients was 280 (200–390) meters. Group 1, 2 and 3 walked 205 (173–328) meters, 285 (200–380) meters and 320 (213–438) meters, respectively. Patients in group 1 desaturated more during the ISWT than group 2 (all p-values <0.001). Furthermore, these patients had a higher rest and maximal dyspnoea Borg score in comparison to group 2 (p<0.001, p = 0.011, respectively). Although the HR before and at the end of the ISWT was comparable between the three groups, the maximal HR during ISWT in ratio to the maximal HR reached during CPET ($HRmax_{ISWT}$/

**Table 1. Subject characteristics, severity of complaints and parameters of pulmonary function and physical performance of the whole group and the three sub-groups based on tolerated duration during the ESWT.**

| Variables | All patients with COPD (n = 245) | Group 1 (n = 41) Tlim <3 min | Group 2 (n = 124) Tlim = 3–8 min | Group 3 (n = 80) Tlim >8 min | p-value |
|---|---|---|---|---|---|
| Gender (male, %) | 114 (47) | 17 (42) | 61 (49) | 36 (45) | 0.653 |
| Age (years) | 61.4 ± 7.8 | 61.9 ± 7.1 | 62.4 ± 7.2 | 59.4 ± 8.6 | 0.021[†] |
| BMI (kg/m$^2$) | 26 ± 6 | 26 ± 7 | 26 ± 5 | 25 ± 6 | 0.195 |
| CCI [a] | 1 (1–2) | 1 (1–3) | 1 (1–2) | 1 (1–2) | 0.853 |
| FFMI [b] | 16.9 ± 2.5 | 16.8 ± 3.1 | 17.2 ± 2.3 | 16.6 ± 2.5 | 0.296 |
| **Severity of complaints** | | | | | |
| mMRC score [c] | 2 (1–3) | 2 (2–3) | 2 (1–3) | 2 (1–3) | 0.006[#,†] |
| CAT score [d] | 17.2 ± 7.0 | 17.3 ± 6.5 | 16.9 ± 7.4 | 17.7 ± 6.7 | 0.784 |
| HADS anxiety score [e] | 8.5 ± 4.2 | 8.7 ± 5.0 | 8.3 ± 4.0 | 8.6 ± 4.1 | 0.852 |
| HADS depression score [e] | 8.6 ± 3.8 | 8.6 ± 4.3 | 8.2 ± 3.7 | 9.1 ± 3.6 | 0.349 |
| CIS fatigue score [f] | 48 (43–53) | 48 (42–52) | 48 (44–53) | 49 (44–54) | 0.836 |
| **Pulmonary function parameters** | | | | | |
| FEV$_1$ (L) [g] | 1.1 (0.8–1.5) | 0.9 (0.7–1.2) | 1.0 (0.8–1.5) | 1.2 (0.9–1.8) | <0.001[*,#] |
| FEV$_1$ (% predicted) [g] | 38 (29–52) | 31 (27–42) | 37 (29–49) | 44 (33–56) | 0.001[#] |
| Tiffeneau index (%) [g] | 34.5 (28.0–45.5) | 31.7 (26.7–38.6) | 33.5 (26.9–43.7) | 37.8 (30.8–50.2) | 0.005[#] |
| FRC (% predicted) [h] | 163 ± 38 | 169 ± 37 | 165 ± 38 | 156 ± 39 | 0.158 |
| RV (% predicted) [i] | 191 ± 53 | 201 ± 52 | 191 ± 54 | 186 ± 53 | 0.394 |
| TLC (% predicted) [i] | 123 ± 19 | 125 ± 20 | 124 ± 19 | 122 ± 18 | 0.591 |
| FRC/TLC (%) [h] | 70 ± 9 | 72 ± 9 | 70 ± 9 | 67 ± 10 | 0.009[#] |
| RV/TLC (%) [i] | 56 ± 10 | 60 ± 9 | 56 ± 10 | 54 ± 10 | 0.026[#] |
| DL$_{CO}$ (mL/mmHg/min) [j] | 3.6 ± 1.6 | 3.0 ± 1.4 | 3.6 ± 1.5 | 3.9 ± 1.8 | 0.011[#] |
| DL$_{CO}$ (% predicted) [j] | 42 ± 16 | 36 ± 15 | 42 ± 15 | 45 ± 18 | 0.021[#] |
| **Physical performance parameters** | | | | | |
| Wmax (Watts) [k] | 70 ± 34 | 53 ± 25 | 68 ± 34 | 82 ± 34 | <0.001[#,†] |
| VO$_2$Max (ml/min/kg) [l] | 13.8 (11.8–16.8) | 12.5 (10.7–13.7) | 13.4 (11.8–16.1) | 16.1 (13.6–18.3) | <0.001[#,†] |
| VO$_2$Max (% predicted) [l] | 58 (48–67) | 52 (45–63) | 57 (47–66) | 61 (50–74) | 0.008[#] |
| CWRT time (s) [m] | 300 (187–495) | 224 (165–290) | 294 (180–433) | 327 (218–600) | 0.028[#] |
| MVC (Nm) [n] | 117 ± 38 | 113 ± 30 | 120 ± 42 | 114 ± 37 | 0.414 |
| MVC (% predicted) [n] | 63 ± 15 | 64 ± 14 | 64 ± 16 | 62 ± 13 | 0.445 |
| Physical activity (steps/day) [o] | 3480 (2386–5168) | 2651 (1517–3923) | 3228 (2388–4752) | 4732 (2934–6097) | <0.001[#,†] |
| Physical activity (average PAL) [p] | 1.34 (1.29–1.42) | 1.30 (1.26–1.35) | 1.34 (1.29–1.41) | 1.38 (1.31–1.48) | <0.001[#,†] |

Data is presented as mean ± SD, median (IQR 25–75%) or number of patients (percentage), as appropriate.

[*] indicates a significant difference after Bonferroni post-hoc correction between group 1 and group 2

[#] indicates a significant difference after Bonferroni post-hoc correction between group 1 and group 3

[†] indicates a significant difference after Bonferroni post-hoc correction between group 2 and group 3. Alphabetic characters in superscript indicate a sample size deviant from n = 245 (group 1: 41, group 2: 124, group 3: 80) with the following: a. n = 226 (37, 116, 73), b. n = 209 (37, 101, 71), c. n = 222 (39, 112, 71), d. n = 214 (37, 106, 71), e. n = 221 (39, 111, 71), f. n = 223 (39, 112, 72), g. n = 240 (41, 121, 78), h. n = 235 (39, 120, 76), i. n = 236 (39, 121, 76), j. n = 228 (39, 114, 75), k. n = 203 (33, 104, 66), l. n = 194 (33, 97, 64), m. n = 154 (21, 78, 55), n. n = 226 (40, 112, 74), o. n = 242 (40, 123, 79), p. n = 241 (40, 122, 79). Definitions of abbreviations: BMI = Body Mass Index, CAT = COPD Assessment Test, CCI = Charlson Comorbidity Index, CIS = Checklist Individual Strength, CWRT = constant work rate cycle test, DL$_{CO}$ = single-breath carbon monoxide diffusion capacity, FEV$_1$ = forced expiratory volume in 1 second, FFMI = fat free mass index, FRC = functional residual capacity, HADS = Hospital Anxiety and Depression Scale, mMRC = modified Medical Research Council, MVC = maximal voluntary contraction, PAL = physical activity level, TLC = total lung capacity, Tlim = tolerated duration, RV = residual volume, VO$_2$max = maximal oxygen uptake, Wmax = maximal workload.

HRmax$_{CPET}$) was lower in patients from group 3 than in patients from group 2 (p = 0.002). Other ISWT performance parameters were not significantly different for group 1 and 3 in comparison to group 2 (Table 2).

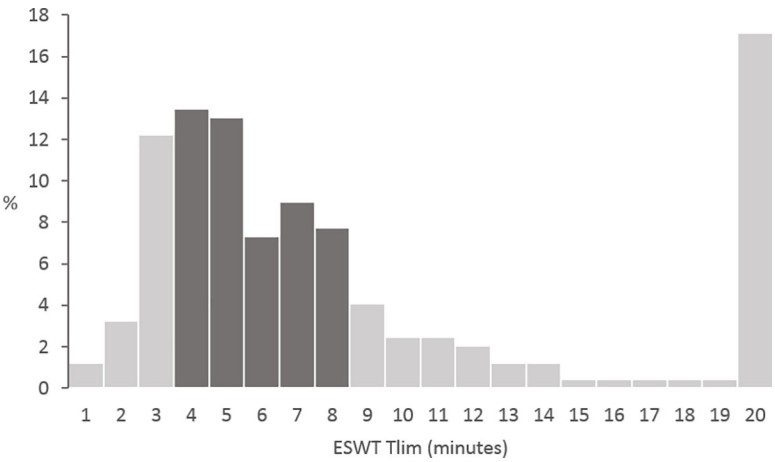

**Fig 2. The distribution of patients according to the ESWT Tlim per minute.**

**ESWT performance parameters.** Patients with an ESWT Tlim >8 minutes had a higher maximal oxygen saturation and higher resting and maximal dyspnoea Borg score during the ESWT than patients from group 2 (all p-values <0.001). Besides speed and time, other ESWT performance parameters of patients from group 1 and 3 were comparable to group 2 (S1 Table).

**Predictors of ESWT time.** Univariate linear regression models that were used to explain ESWT Tlim variability are documented in S2 and S3 Tables. Age, BMI, mMRC and several variables of pulmonary function ($FEV_1$, Tiffeneau index, FRC, FRC/TLC, RV/TLC and $DL_{CO}$), physical performance (Wmax, $VO_2$max, CWRT time, steps/day and average PAL) and ISWT performance (distance, speed, $SpO_2$, $HRmax_{ISWT}$/$HRmax_{CPET}$ and Borg scores for dyspnoea and leg fatigue) were significant explanatory variables of ESWT Tlim in univariate analyses.

In a multivariate linear regression model, BMI, $VO_2$max, CWRT time, average PAL, ISWT speed, dyspnoea Borg score at rest and increase of leg fatigue Borg score during ISWT were independent predictors of ESWT Tlim (S4 Table). This model explained ~30% of the variability in ESWT Tlim ($R^2 = 0.297$, p<0.001).

## Discussion

The current study confirmed that patients with COPD display a large variability in ESWT Tlim, even though ESWT was performed at a fixed percentage of pre-determined maximal walking speed. To our knowledge, this is the first study that determined possible predictors of ESWT Tlim in patients with COPD. We found that BMI, $VO_2$max, CWRT time, average PAL, ISWT speed, dyspnoea Borg score at rest and increase of leg fatigue Borg score during ISWT are independent predictors of ESWT Tlim. However, collectively these determinants can only explain ~30% of ESWT Tlim variability.

### ESWT Tlim highly variable

A large interindividual variability in ESWT Tlim was illustrated by the notion that half of the patients with COPD performed the ESWT outside the desired duration of 3–8 minutes. Furthermore, a required termination of the test was needed in 17% of the patients as they reached the maximum test duration of 20 minutes, but probably could have walked even longer. This

**Table 2. ISWT performance parameters of the whole group and the three subgroups based on tolerated duration during the ESWT.**

| Variables | All patients with COPD (n = 245) | Group 1 (n = 41) Tlim <3 min | Group 2 (n = 124) Tlim = 3–8 min | Group 3 (n = 80) Tlim >8 min | p-value |
|---|---|---|---|---|---|
| Distance (m) [a] | 280 (200–390) | 205 (173–328) | 285 (200–380) | 320 (213–438) | 0.004[#] |
| Distance (% predicted) [b] | 45 (30–58) | 33 (25–49) | 47 (29–58) | 47 (34–64) | 0.015[#] |
| Speed (km/h) | 4.8 (4.2–5.4) | 4.2 (3.6–5.1) | 4.8 (4.2–5.4) | 4.8 (4.2–6.0) | 0.029[#] |
| $SpO_2$ rest (%) | 96 (94–97) | 95 (93–96) | 96 (94–97) | 96 (94–98) | 0.146 |
| $SpO_2$ at max (%) | 89 (85–94) | 85 (82–91) | 90 (85–94) | 90 (87–94) | 0.001[*,#] |
| $SpO_2$ delta (max-rest, %) | -7 (-11- -2) | -9 (-13- -6) | -6 (-11- -2) | -5 (-9- -1) | 0.002[*,#] |
| HR rest (bpm) [c] | 84 ± 12 | 86 ± 12 | 84 ± 13 | 83 ± 10 | 0.632 |
| HR at max (bpm) [c] | 113 ± 19 | 114 ± 16 | 114 ± 21 | 110 ± 17 | 0.507 |
| HRmax$_{ISWT}$/HRmax$_{CPET}$ [d] | 92 ± 14 | 97 ± 10 | 94 ± 17 | 87 ± 11 | 0.002[#,†] |
| HR delta (max-rest, bpm) [c] | 29 ± 15 | 27 ± 11 | 30 ± 17 | 27 ± 15 | 0.379 |
| Borg score dyspnoea rest | 2 (1–3) | 2 (1–3) | 2 (1–3) | 1 (0–2) | 0.058 |
| Borg score dyspnoea max | 5 (4–7) | 7 (5–7) | 5 (4–7) | 5 (3–7) | <0.001[*,#] |
| Borg score dyspnoea delta | 3 (2–5) | 5 (3–6) | 3 (2–5) | 3 (2–5) | 0.011[*,#] |
| Borg score fatigue rest | 2 (1–3) | 3 (1–5) | 2 (1–3) | 2 (1–4) | 0.221 |
| Borg score leg fatigue max | 5 (3–7) | 5 (3–7) | 5 (3–7) | 4 (3–6) | 0.238 |
| Borg score leg fatigue delta | 2 (1–4) | 3 (1–5) | 2 (1–4) | 2 (0–3) | 0.065 |

Data is presented as mean ± SD or median (IQR 25–75%), as appropriate.

* indicates a significant difference after Bonferroni post-hoc correction between group 1 and group 2

# indicates a significant difference after Bonferroni post-hoc correction between group 1 and group 3

† indicates a significant difference after Bonferroni post-hoc correction between group 2 and group 3. Alphabetic characters in superscript indicate a sample size deviant from n = 245 (group 1: 41, group 2: 124, group 3: 80) with the following: a. n = 242 (40, 122, 80), b. n = 243 (40, 123, 80), c. n = 184 (27, 91, 66), d. n = 167 (25, 80, 62).

Definitions of abbreviations: HR = heartrate, HRmax$_{ISWT}$/HRmax$_{CPET}$ = maximal HR of the incremental shuttle walk test relative to the maximal HR during the cardiopulmonary exercise test, $SpO_2$ = peripheral capillary oxygen saturation, Tlim = tolerated duration.

large interindividual variability is in accordance with a recent study of Maltais et al., who investigated the responsiveness of the ESWT to bronchodilation [13]. In their analysis the authors were urged to exclude patients that had a baseline ESWT Tlim of more than 15 minutes to allow measurable room for improvement on a post-intervention ESWT. So, high interindividual variability in ESWT Tlim requires a larger number of participants in clinical studies to detect an effect of interventions. The interpretation of intervention efficacy is even more complicated as the potential effect size (i.e. post–pre intervention) of ESWT Tlim depends on the pre-intervention ESWT Tlim due to the hyperbolic nature of the load-duration relationship [12]. Therefore, in a population with a high interindividual variability in ESWT Tlim, individual effects of interventions are difficult to compare [12, 14, 15]. The best solution to reduce this variability is to determine the load-duration relationship in every individual. However, this is clinically impractical because it requires the completion of several ESWT tests at various intensities. Another possibility is to perform a second ESWT at an adjusted pace in patients with an ESWT Tlim <3 or >8 minutes. However, the size of the adjustment in pace has not been determined yet and a second ESWT is not always possible due to practical reasons like time constraints. Therefore, it is important to search for other possibilities to reduce the variability in ESWT Tlim. One option would be to better predict the ESWT Tlim prior to its performance in order to individually adjust the ESWT pace with clinical available measures. Therefore, this study further investigated correlates of ESWT Tlim variability.

## Pulmonary function and physical performance parameters

Patients performing the ESWT longer than 8 minutes had a higher physical capacity and activity in comparison to group 2. In addition, exercise tolerance obtained by CWRT was also positively related to the ESWT Tlim. On the other hand, patients that could not sustain the ESWT for at least 3 minutes, were characterized by a lower $FEV_1$ (L) than patients in group 2 and several pulmonary function measures were negatively associated with ESWT Tlim. Pulmonary dysfunction is a well-known contributor to exercise intolerance in patients with COPD [10, 11]. In short, ventilatory capacity is limited by airflow obstruction and hyperinflation, which may even exacerbate during exercise. On the other hand, ventilatory demand in patients with COPD may be increased as a result of abnormal pulmonary gas exchange, increased work of the respiratory muscles and early lactate production in the peripheral muscles [16, 37–40]. This leads to increased sensations of dyspnoea during exercise and explains why the severity of pulmonary dysfunction is related to the ability to sustain a certain exercise load [37, 38]. However, it should be stressed that the load of the ESWT is normalized for maximal exercise capacity, as ESWT pace is individually set at 85% of maximal ISWT pace. Despite this normalization, pulmonary function, physical capacity and physical activity are still related to the time patients can sustain this individually assessed pace. This suggests that the load-duration relationship is affected by these measures. The load-duration relation is described by two parameters; the critical load and the curvature constant [10–12]. Neder *et al.* previously reported that both parameters are reduced in patients with COPD in comparison to healthy controls [11]. The results of the current study suggest that even within the COPD population, critical load and the curvature constant might be influenced by pulmonary function, physical capacity and physical activity.

Thus, the current findings suggest that, in addition to maximal ISWT pace, measures of pulmonary function, physical capacity and physical activity, if clinically available, might be helpful to more adequately set ESWT pace. It appears that patients with more severe airway obstruction should be set at paces slower than 85% of maximal pace and patients with higher physical capacity and activity levels at paces faster than 85% of maximal pace. However, exact cut-off values and sizes of adjustment should be explored in future studies.

## ISWT performance parameters

Because ESWT pace is based on maximal ISWT speed, it is essential that the ISWT is performed with maximal effort and using a standardized operating procedure. Therefore, we investigated performance measures obtained during ISWT. In contrast to healthy individuals, the exercise capacity of most patients with COPD is not limited by cardiac output [37]. So maximum HR during ISWT cannot be used to establish maximal effort in patient with COPD. Accordingly, we examined ISWT HR in ratio to the maximum HR obtained during CPET, i.e. $HRmax_{ISWT}/HRmax_{CPET}$. We found that this ratio was significantly lower in patients that displayed an ESWT Tlim >8 minutes (group 3). Furthermore, the ratio was negatively associated with ESWT Tlim. This indicates that some patients with an ESWT time >8 minutes might have performed sub-maximally on their ISWT. A longer ESWT Tlim was also associated with a reduced increase of perceived leg fatigue during ISWT. Although this is a subjective measure, it is in line with a sub-maximal effort during the ISWT. A more objective indication of maximal effort can be provided by additional physiological measures like minute ventilation, oxygen consumption and/or blood lactate values [31], but these would make the ISWT less accessible and more expensive. Our data suggest that if maximal attained HR during CPET is available it can be used together with simple non-invasive measurement of HR during ISWT to provide an estimation of ISWT effort.

Additionally, patients with a shorter ESWT Tlim desaturated more during the ISWT than patients with a longer ESWT Tlim, which was also reported during the ESWT. This might suggest that patients who desaturate more during the ISWT should perform the ESWT at a lower relative load than 85%.

### Predicting ESWT

Because our data showed that several clinically obtained measures significantly correlated with ESWT Tlim, we further investigated if a model could be built to predict ESWT Tlim. Based on a multivariate linear regression model, BMI, $VO_2$max, CWRT time, physical activity, ISWT speed, dyspnoea Borg score at rest and delta leg fatigue Borg score during ISWT were identified as independent predictors of ESWT Tlim. Although each of these parameters significantly contributes to ESWT Tlim variability, the total explained variance is only ~30%. Therefore, it is important to evaluate additional factors that might be associated with ESWT Tlim. For example, dynamic hyperinflation and reduced leg muscle endurance are known to frequently occur in patients with COPD and affect exercise tolerance independent of the severity of pulmonary dysfunction [2, 39, 41–43]. Further research is necessary to assess if these or other factors could improve the accuracy of predicting ESWT Tlim. Eventually, a proper prediction model with clinical available measures might help clinicians to identify patients that are expected to reach an ESWT Tlim outside the desired timeframe of 3–8 minutes [44] and to decide if ESWT pace should be set a different level than 85% of maximal ISWT pace.

### Study limitations

In our design ESWT pace was based on patients' maximal pace obtained from one ISWT. Because Dyer et al. reported a learning effect in a second ISWT [45], two ISWT tests are recommended when the ISWT is used to measure change over time or interventions [19]. However, when ISWT is only used to set ESWT pace, one test has been postulated to be sufficient [46]. Although we expect that performing two ISWT's prior to ESWT might reduce the inter-individual variability, it does not necessarily eliminate the need of a second ESWT in all patients. Furthermore, two ISWT's would increase the amount of tests in all patients, while our data show that with one ISWT half of the patients perform their ESWT within the desired duration of 3–8 minutes.

## Conclusion

This study confirmed a large interindividual variability in ESWT Tlim in patients with COPD, as only half of the patients reached an ESWT Tlim within the desired duration of 3–8 minutes. Our data showed that next to maximal ISWT speed, other ISWT performance measures as well as clinically available measures of pulmonary function, physical capacity and physical activity were independent determinants of ESWT Tlim. Nevertheless, these determinants could only explain ~30% of its variability. Therefore, future studies are needed to establish whether these and additional factors can be used to better adjust individual ESWT pace in order to reduce ESWT Tlim variability.

## Supporting information

**S1 Table. ESWT parameters of the whole group and the three subgroups based on tolerated duration during the ESWT.**
(PDF)

**S2 Table. Univariate linear regression models for the subject characteristics, severity of complaints, pulmonary function and physical performance with the tolerated duration on the ESWT.**
(PDF)

**S3 Table. Univariate linear regression models for the ISWT parameters with the tolerated duration on the ESWT.**
(PDF)

**S4 Table. Multivariate linear regression analysis to predict tolerated duration on the ESWT.**
(PDF)

## Acknowledgments

We would like to thank A.R.T. Donders (Radboudumc) for his input in the statistical analyses and interpretation and W. Derave (Ghent University) for his input and collaboration within the BASES consortium, in the context of which the current manuscript was written. The BASES consortium consists of M.A. Spruit (CIRO, lead author, e-mail: martijnspruit@ciro-horn.nl), A.A.F. Stoffels (Radboudumc), R. Meys (CIRO), P. Klijn (Merem), H.W.H. van Hees (Radboudumc), C. Burtin (Hasselt University), F.M.E. Franssen (CIRO), B. van den Borst (Radboudumc), J. De Brandt (Hasselt University), M.J.H. Sillen (CIRO), E.F.M. Wouters (CIRO), E. bij de Vaate (Merem), F.N. Schleich (CHU Sart-Tilman Liege), M. Hayot (University of Montpellier–Montpellier CHU), P. Pomiès (University of Montpellier–Montpellier CHU), W. Derave (Ghent University) and I. Everaert (Ghent University).

## Author Contributions

**Conceptualization:** Anouk A. F. Stoffels, Hieronymus W. H. van Hees.

**Data curation:** Anouk A. F. Stoffels, Jeannette B. Peters.

**Formal analysis:** Anouk A. F. Stoffels.

**Funding acquisition:** Martijn A. Spruit.

**Investigation:** Anouk A. F. Stoffels, Mariska P. M. Klaassen, Hanneke A. C. van Helvoort.

**Methodology:** Anouk A. F. Stoffels, Bram van den Borst, Hieronymus W. H. van Hees.

**Project administration:** Anouk A. F. Stoffels.

**Supervision:** Bram van den Borst, Hieronymus W. H. van Hees.

**Visualization:** Anouk A. F. Stoffels.

**Writing – original draft:** Anouk A. F. Stoffels, Hieronymus W. H. van Hees.

**Writing – review & editing:** Anouk A. F. Stoffels, Bram van den Borst, Jeannette B. Peters, Mariska P. M. Klaassen, Hanneke A. C. van Helvoort, Roy Meys, Peter Klijn, Chris Burtin, Frits M. E. Franssen, Alex J. van 't Hul, Martijn A. Spruit, Hieronymus W. H. van Hees.

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
