## [Decision Letter · Decision Letter 0]

12 Feb 2021

PONE-D-20-32526

Correlates of variability in endurance shuttle walk test time in patients with chronic obstructive pulmonary disease

PLOS ONE

Dear Dr. Stoffels,

Thank you for submitting your manuscript to PLOS ONE and for your patience on this review. After careful consideration, we feel that it has merit but does not fully meet PLOS ONE’s publication criteria as it currently stands. Therefore, we invite you to submit a revised version of the manuscript that addresses the points raised during the review process. To facilitate your revisions, I have outline some key areas to address, in addition to our reviewers. These include:

Following your purpose statement in the intro, add clearly identified hypotheses.Please reword the conclusion statements for the abstract and paper to the past tense.Be sure to proof read for consistent use of formatting.I agree with the reviewer. A flow chart of some type would aid for the inclusion/exclusion section.Likewise, please include effect sizes.Suggestion: Line 231, change high variability (vague) to **ESWT Tlim highly variable**As noted above, much of the conclusion is written in present tense.

We look forward to receiving your revised manuscript.

Kind regards,

Chris Harnish, PhD

Academic Editor

PLOS ONE

Journal Requirements:

2. In the methods section please provide a reference to the main study from which the secondary analysis was performed from.

"Dr. F.M.E. Franssen is supported by grants and personal fees from AstraZeneca, personal fees from Boehringer Ingelheim, personal fees from Chiesi, personal fees from GlaxoSmithKline, grants and personal fees from Novartis, personal fees from TEVA, outside the submitted work. Dr. B. van den Borst is supported by personal lecture fees from AstraZeneca and Boehringer Ingelheim bv.

A.A.F. Stoffels, R. Meys, H.W.H. van Hees, P. Klijn, C. Burtin, M.A. Spruit, H.A.C. van Helvoort, J.B. Peters, M.P.M. Klaassen and A.J. van ‘t Hul declare that they do not have a conflict of interest."

5. One of the noted authors is a group or consortium [BASES consortium]. In addition to naming the author group, please list the individual authors and affiliations within this group in the acknowledgments section of your manuscript. Please also indicate clearly a lead author for this group along with a contact email address.

Reviewers' comments:

Reviewer's Responses to Questions

**Comments to the Author**

1. Is the manuscript technically sound, and do the data support the conclusions?

Reviewer #1: Yes

2. Has the statistical analysis been performed appropriately and rigorously? 

Reviewer #1: Yes

3. Have the authors made all data underlying the findings in their manuscript fully available?

Reviewer #1: No

4. Is the manuscript presented in an intelligible fashion and written in standard English?

Reviewer #1: Yes

5. Review Comments to the Author

Reviewer #1: General Comments

The reviewer would like to thank the authors for taking the time to produce a well described and written study. Please see specific comments below.

Abstract

Line 28: Suggest changing to not completely understood.

Line 32: If there is room, please include inclusion/exclusion criterion

Line 35: Please add statistical analyses used.

Line 43: suggest moving to past tense: Secondly, these results demonstrated …

Introduction

Overall the introduction is well defined and written.

Line 67: Like within the abstract, suggest changing to not completely understood.

Methods

Line 114: nine meters, instead of nine meter. (same with 10 meters).

Line 124: percentage of predicted…please be more specific of the predicted metric.

Line 127: Paragraphs starting here are indented while all others beforehand are not. Please revise.

Line 124-126: Suggest having a separate paragraph prior to test that describe inclusion and exclusion criterion. Further a flow chart may aid the reader in understanding how the sample was obtained.

Statistical analyses: Suggest including an effect size index analysis for significant difference findings as this will give further context of the magnitude of these potential differences.

Results

Well described

Discussion

Overall well done .

6. PLOS authors have the option to publish the peer review history of their article (what does this mean?). If published, this will include your full peer review and any attached files.

Reviewer #1: **Yes: **Garrett Scott Bullock

---

## [Author Response · Author response to Decision Letter 0]

19 Mar 2021

Response to reviewers

We would like to thank the editor and the reviewer for their constructive comments and giving us the opportunity to revise our manuscript. We provided a point-by point reply below and adapted the manuscript accordingly.

*The reported page and line numbers correspond to the manuscript with track changes (view: all markup).

Comments editor

C1. Following your purpose statement in the intro, add clearly identified hypotheses.

R1. We have added our hypothesis after the aim in the introduction: “A priori, we hypothesized that parameters of pulmonary function and physical performance are independent determinants of ESWT Tlim and can partly explain the high variability of ESWT Tlim in patients with COPD”, see page 6, lines 102-105. 

C2. Please reword the conclusion statements for the abstract and paper to the past tense.

R2. The conclusion statement in the abstract and paper has been changed to the past tense, see page 4, lines 54-60 (abstract) and pages 20-21, lines 401-408 (paper). 

C3. Be sure to proof read for consistent use of formatting.

R3. We have proof read the article and made some changes to obtain a consistent format of the manuscript. Changes are marked in the tracked changes version of the revised manuscript.

C4. I agree with the reviewer. A flow chart of some type would aid for the inclusion/exclusion section.

R4. We agree with the reviewer and made a flow chart that shows how the number of patients used for the analyses was reached, see figure 1.

C5. Likewise, please include effect sizes.

R5. Thank you for your suggestion. We assume that the reviewer suggests to include a column in tabel1, 2 and S2 with standardized effect sizes on differences of the parameters described. We have discussed this topic with our statistician (dr. A.R.T. Donders). In our opinion, the interpretation of the magnitude of differences will not be improved by including such effect size numbers, because 1) effect sizes of 2 or 3 comparisons per parameter are difficult to reflect in one number and 2) both non-parametric and parametric testing was applied, which affects the interpretation of effect sizes. So, we think that the raw data as presented in table 1,2 and S2 is the most appropriate way to show the magnitude of the differences across the three groups. Nevertheless, as we reviewed table 1, we noted that the difference in mMRC score may not be clear to the readers. In the revised manuscript we therefore expressed the mean mMRC score in the text as well ‘Patients in group 3 were younger (59.4 ± 8.6 years) than patients in group 2 (62.4 ± 7.2 years, p=0.021). The severity of dyspnea sensation, as reflected by the mMRC score, was lower in patients from group 3 (median 2(1-3) and mean 1.8±1.2) than in patients from group 2 (median 2(1-3) and mean 2.2±1.2, p=0.006).’., see page 11, lines 214-216.

C6. Suggestion: Line 231, change high variability (vague) to ESWT Tlim highly variable

R6. Thank you for this suggestion, it has been changed to ‘ESWT Tlim highly variable’ in the manuscript on page 16, line 297. 

C7. As noted above, much of the conclusion is written in present tense.

R7. The conclusion has been rewritten in the past tense, see pages 20-21, lines 401-408.

‘This study confirmed a large interindividual variability in ESWT Tlim in patients with COPD, as only half of the patients reached an ESWT Tlim within the desired duration of 3-8 minutes. Our data showed that next to maximal ISWT speed, other ISWT performance measures as well as clinically available measures of pulmonary function, physical capacity and physical activity were independent determinants of ESWT Tlim. Nevertheless, these determinants could only explain ~30% of its variability. Therefore, future studies are needed to establish whether these and additional factors can be used to better adjust individual ESWT pace in order to reduce ESWT Tlim variability.’

C8. Please ensure that your manuscript meets PLOS ONE’s style requirements, including those for file naming. The PLOS ONE style templates can be found at: https://journals.plos.org/plosone/s/file?id=wjVg/PLOSOne_formatting_sample_main_body.pdf and https://journals.plos.org/plosone/s/file?id=ba62/PLOSOne_formatting_sample_title_authors_affiliations.pdf

R8. The manuscript has been checked and adapted to the PLOS ONE requirements. 

C9. In the methods section please provide a reference to the main study from which the secondary analysis was performed form.

R9. The data is obtained as part of standard care during baseline assessment of a pulmonary rehabilitation program in our center. There is no main study from which this is a secondary analysis. However, as we understand the confusion, the term ‘standard care’ has been added to the methods section, ‘The data was collected during baseline assessment as part of standard care of the PR program.’ see page 7, lines 112-113. 

C10. We note that the grant information you provided in the ‘Funding information’ and ‘Financial Disclosure’ sections do not match. When you resubmit, please ensure that you provide the correct grant numbers for the awards you received for the study in the ‘Funding Information’ section.

R10. Thank you for noticing. The financial disclosure should be changed to ‘The BASES consortium is financially supported by the Lung Foundation, the Netherlands (#5.1.18.232).’ as mentioned in the cover letter.

The mentioned grants of F. Franssen and B. van den Borst have no influence and are unrelated to the submitted article. Furthermore, this information has already been mentioned in the ‘conflicts of interests’ section in the manuscript. We apologize for the confusion and our mistake. As we cannot change the financial disclosure section ourselves, we would appreciate if you can change this for us. If you have further questions regarding the funding, please contact us.

Initially, we did not provide a name for recipient of the grant of the Lung foundation as this has been provided to the BASES consortium. We have changed this to Martijn Spruit as he is the lead author of the BASES consortium. 

C11. Thank you for stating the following in the Competing Interests section:

"Dr. F.M.E. Franssen is supported by grants and personal fees from AstraZeneca, personal fees from Boehringer Ingelheim, personal fees from Chiesi, personal fees from GlaxoSmithKline, grants and personal fees from Novartis, personal fees from TEVA, outside the submitted work. Dr. B. van den Borst is supported by personal lecture fees from AstraZeneca and Boehringer Ingelheim bv. A.A.F. Stoffels, R. Meys, H.W.H. van Hees, P. Klijn, C. Burtin, M.A. Spruit, H.A.C. van Helvoort, J.B. Peters, M.P.M. Klaassen and A.J. van ‘t Hul declare that they do not have a conflict of interest."

R11. We have added the statement to the ‘conflicts of interest’ section, see page 23, line 439. 

C12. One of the noted authors is a group or consortium [BASES consortium]. In addition to naming the author group, please list the individual authors and affiliations within this group in the acknowledgements section of your manuscript. Please also indicate clearly a lead author for this group along with a contact email address. 

R12. We have listed the individual authors of the BASES consortium to the acknowledgements section, including the affiliations, see page 22, lines 415-421. 

‘The BASES consortium consists of M.A. Spruit (CIRO, lead author, e-mail: martijnspruit@ciro-horn.nl), A.A.F. Stoffels (Radboudumc), R. Meys (CIRO), P. Klijn (Merem), H.W.H. van Hees (Radboudumc), C. Burtin (Hasselt University), F.M.E. Franssen (CIRO), B. van den Borst (Radboudumc), J. De Brandt (Hasselt University), M.J.H. Sillen (CIRO), E.F.M. Wouters (CIRO), E. bij de Vaate (Merem), F.N. Schleich (CHU Sart-Tilman Liege), M. Hayot (University of Montpellier – Montpellier CHU), P. Pomiès (University of Montpellier – Montpellier CHU), W. Derave (Ghent University) and I. Everaert (Ghent University).’

C13. Please include captions for you Supporting Information files at the end of your manuscript, and update any in-text citations to match accordingly. Please see our Supporting Information Guidelines for more information: http://journals.plos.org/plosone/s/supporting-information.

R13. We have added a Supporting Information Caption to the manuscript, see page 23, lines 441-445.

‘S1 Table. ESWT parameters for the whole group and subgroups. 

S2 Table. Univariate linear regression models for the clinical parameters and ESWT Tlim.

S3 Table. Univariate linear regression models for the ISWT parameters and ESWT Tlim

S4 Table. Multivariate linear regression analysis to predict ESWT Tlim.’

 

Reviewer 1

C1. Abstract: line 28: Suggest changing to not completely understood.

R1. We agree with this suggestion and it has been changed to ‘.. factors determining ESWT Tlim are not completely understood.’ in the manuscript on page 3, line 35. 

C2. Abstract: line 32: If there is room, please include inclusion/exclusion criterion

R2. The inclusion criteria have been added to the abstract: ‘Inclusion criteria were: diagnosis of COPD and complete data availability regarding ESWT and ISWT.’ see page 3, lines 39-40. 

C3. Abstract: line 35: Please add statistical analyses used.

R3. Thank you for this suggestion. We understand the additional value of describing the statistical analyses in the abstract. However, the maximal number of words does not allow us to describe the statistical analyses clearly and complete. Therefore, we have chosen to leave the statistical analysis out of the abstract. 

C4. Abstract: line 43: suggest moving to past tense: Secondly, these results demonstrated..

R4. Thank you for the suggestion, the conclusion has been changed to the past tense, see page 4, lines 54-60. 

‘Conclusion: This study reported a large variability in ESWT Tlim in COPD patients. Secondly, these results demonstrated that next to maximal ISWT speed, other ISWT performance measures as well as clinical measures of pulmonary function, physical capacity and physical activity were independent determinants of ESWT Tlim. Nevertheless, as these determinants only explained ~30% of the variability, future studies are needed to establish whether additional factors can be used to better adjust individual ESWT pace in order to reduce ESWT Tlim variability.’ 

C5. Introduction: Overall the introduction is well defined and written.

Line 67: Like within the abstract, suggest changing to not completely understood.

R5. Thank you for the suggestion, this was adjusted in the manuscript on page 5, line 84. 

C6. Methods: line 114: nine meters, instead of nine meter. (same with 10 meters).

R6. Thank you for the suggestion, it has been changed in the manuscript to ‘The ISWT required the patients to walk around two markers set nine meters apart (10 meters course) …’ on page 8, lines 146-147. 

C7. Methods: line 124: percentage of predicted…please be more specific of the predicted metric.

R7. We have added ‘in meters’ after ISWT distance to clarify the used metric ‘the ISWT distance in meters was calculated as percentage of predicted’, see page 9, lines 155-156. 

C8. Methods: line 127: Paragraphs starting here are indented while all others beforehand are not. Please revise.

R8. We have indented all paragraphs. 

C9. Methods: line 124-126: Suggest having a separate paragraph prior to test that describe inclusion and exclusion criterion. Further a flow chart may aid the reader in understanding how the sample was obtained.

R9. We agree that this will help the reader to understand how the sample was obtained, so we added a flowchart with in- and exclusion criteria to the manuscript, see figure 1.

C10. Methods: statistical analyses: Suggest including an effect size index analysis for significant difference findings as this will give further context of the magnitude of these potential differences.

R10. Thank you for your suggestion. If we understand correctly, you suggest to include a

column in tabel1, 2 and S2 with standardized effect sizes on differences of the parameters 

described in those tables. We have discussed this topic with our statistician (dr. A.R.T. 

Donders). We think that the interpretation of such effect size is hampered, because 1) 

effect sizes of 2 or 3 comparisons per parameter are difficult to reflect in one number and 

2) both non-parametric and parametric testing was applied. So, in our opinion the raw 

data as presented in table 1,2 and S2 is the most appropriate way to show the magnitude 

of the differences across the three groups. Nevertheless, as we reviewed table 1, we noted that the difference in mMRC score may not be clear to the readers. In the revised manuscript we therefore expressed the mean mMRC score in the text as well ‘Patients in group 3 were younger (59.4 ± 8.6 years) than patients in group 2 (62.4 ± 7.2 years, p=0.021). The severity of dyspnea sensation, as reflected by the mMRC score, was lower in patients from group 3 (median 2(1-3) and mean 1.8±1.2) than in patients from group 2 (median 2(1-3) and mean 2.2±1.2, p=0.006).’., see page 11, lines 214-216.

C11. Results: well described.

C11. Thank you!

C12. Discussion: overall well done.

C12. Thank you very much for your constructive comments!

---

## [Decision Letter · Decision Letter 1]

25 Mar 2021

Correlates of variability in endurance shuttle walk test time in patients with chronic obstructive pulmonary disease

PONE-D-20-32526R1

Dear Dr. Stoffels,

We’re pleased to inform you that your manuscript has been judged scientifically suitable for publication and will be formally accepted for publication once it meets all outstanding technical requirements.

Kind regards,

Chris Harnish, PhD

Academic Editor

PLOS ONE

Additional Editor Comments (optional):

Reviewers' comments:

Reviewer's Responses to Questions

**Comments to the Author**

1. If the authors have adequately addressed your comments raised in a previous round of review and you feel that this manuscript is now acceptable for publication, you may indicate that here to bypass the “Comments to the Author” section, enter your conflict of interest statement in the “Confidential to Editor” section, and submit your "Accept" recommendation.

Reviewer #1: All comments have been addressed

2. Is the manuscript technically sound, and do the data support the conclusions?

Reviewer #1: Yes

3. Has the statistical analysis been performed appropriately and rigorously? 

Reviewer #1: Yes

4. Have the authors made all data underlying the findings in their manuscript fully available?

Reviewer #1: Yes

5. Is the manuscript presented in an intelligible fashion and written in standard English?

Reviewer #1: Yes

6. Review Comments to the Author

Reviewer #1: The addressed the reviewer comments. I have no other editorial comments. I commend the authors on their work.

7. PLOS authors have the option to publish the peer review history of their article (what does this mean?). If published, this will include your full peer review and any attached files.

Reviewer #1: **Yes: **Garrett Scott Bullock

---

## [Editor Report · Acceptance letter]

12 Apr 2021

PONE-D-20-32526R1 

Correlates of variability in endurance shuttle walk test time in patients with chronic obstructive pulmonary disease 

Dear Dr. Stoffels:

I'm pleased to inform you that your manuscript has been deemed suitable for publication in PLOS ONE. Congratulations! Your manuscript is now with our production department. 

Kind regards, 

on behalf of

Dr. Chris Harnish 

Academic Editor

PLOS ONE